# Ultrasound-Treated and Thermal-Pasteurized Hawthorn Vinegar: Antioxidant and Lipid Profiles in Rats

**DOI:** 10.3390/nu15183933

**Published:** 2023-09-11

**Authors:** Deniz Karakçı, Buket Bakır, Nilay Seyidoglu, Seydi Yıkmış

**Affiliations:** 1Department of Biochemistry, Faculty of Veterinary Medicine, Tekirdag Namik Kemal University, Tekirdag 59030, Turkey; 2Department of Histology and Embryology, Faculty of Veterinary Medicine, Tekirdag Namik Kemal University, Tekirdag 59030, Turkey; buketbakir@nku.edu.tr; 3Department of Physiology, Faculty of Veterinary Medicine, Tekirdag Namik Kemal University, Tekirdag 59030, Turkey; nseyidoglu@nku.edu.tr; 4Department of Food Technology, Tekirdag Namik Kemal University, Tekirdag 59860, Turkey; syimis@nku.edu.tr

**Keywords:** antioxidant, hawthorn, rat, ultrasound

## Abstract

The hawthorn fruit, a member of the Rosaceae family, is a medicinal plant with numerous therapeutic properties. It has a wide range of variants, with *Crataegus tanacetifolia* being the most widely recognized species in the world. The hawthorn fruit has various biological activities, including anti-inflammatory, antibacterial, antioxidant, immune-modulating, and anti-carcinogenic properties. This study focused on improving the antioxidant activity of hawthorn vinegar via different methods. We also aimed to investigate the influence of its hepatic antioxidant abilities on health and extend the shelf life of the vinegar. In the study, the vinegar was produced from the hawthorn fruit, and thermal pasteurization and ultrasound techniques were applied. A total of 56 female adult Wistar-Albino rats were allocated into seven groups and administered hawthorn fruit vinegar via oral gavage on a daily basis. The experimental groups included rats treated with pasteurized vinegar (HVP), ultrasound-treated rats (HVU), and an untreated group that received regular vinegar (HVN) at two different dosage levels (0.5 and 1 mL/kg). The SOD, MDA, and CAT antioxidant levels were measured using the ELISA method in plasma and liver tissue samples. The total plasma cholesterol, triglyceride, HDL, LDL, AST, and ALT values were quantified using commercially available kits. The levels of SOD and CAT in the plasma and liver were found to be significantly higher in the HVU1 group compared to all other groups. Furthermore, the HVU1 cohort exhibited the highest HDL value in plasma. The plasma LDL levels were comparably low in both the thermal-pasteurized and ultrasound-treated groups. There were significant expressions of both CAT and SOD in the liver tissues of the HVU groups (analyzed immunohistochemically). These results indicated that hawthorn vinegar administration with 1 mL/kg in group HVU1 could significantly enhance antioxidant capacity in the liver and, consequently, overall health. It can be suggested that the possible therapeutic effects of hawthorn vinegar may boost its antioxidant capabilities and contribute to an overall improvement in quality of life.

## 1. Introduction

Hawthorn is a popular fruit that has also been labeled as a safe and traditional medicine by the European Medicine Agency [1,2]. The hawthorn plant is a leafy and thorny plant belonging to the Rosaceae family and the Maloideae subfamily [3]. It was also estimated that hawthorn species (*Crataegus* spp.) include 150 to 1200 species. Also, it has been reported that hawthorn fruit is rich in flavonoids, anthociyanaidin, vitamin C, and natural antioxidants [4]. In addition, some of the essential components of hawthorn, namely organic acids, soma amins, and proanthocyanidins, were determined by researchers [5].

Hawthorn fruit is an economically crucial seasonal fruit that is produced in various countries in the world [6]. Hawthorn species (*Crataegus* spp.) were commonly consumed in North America, Europe, and Asia worldwide, as well as Turkey, and have been used as leaves, flowers, and fruits over the years [7]. Hawthorn is a readily accessible fruit purchasable in Turkey, and it is commonly available at a variety of retail establishments that cater to mass consumer markets. Although hawthorn fruit has numerous varieties locally, the most popular species is *Crataegus tanacetifolia* in Turkey, which we also used in this recent study.

Hawthorn fruit has become an important component of nutrition due to its various beneficial properties, such as its ability to help reduce blood pressure, regulate blood glucose and lipid metabolism [8], and decrease the risk of diabetes mellitus and cardiovascular diseases [9]. Furthermore, it is an antioxidant superfood because of its rich flavonoid and polyphenolic compound contents. Data suggest that this fruit could show anti-inflammatory [10], antibacterial [11], antioxidant [12], immune-modulator [13], and anti-cancerogenic [4] activity. The hawthorn fruit’s total phenolic and flavonoid compounds are assumed to be directly proportional to their number of free radicals scavengers [14]. Wang et al. [15] studied hawthorn fruit extract with senescence-accelerated mice. They reported that while SOD (Superoxide dismutase), CAT (Catalase), and GPx (Glutathione peroxidase) antioxidant parameters increased in the plasma, liver, and brain in the hawthorn group, MDA decreased lipid oxidation. Based on this result, it was concluded that the possible antioxidant mechanism of the hawthorn plant in vivo was realized by increasing the number of endogenous antioxidant enzymes. The formation of many diseases, such as cancer, Alzheimer’s, and diabetes mellitus, happens with an excessive increase in free radicals. After oxidative damage, unaccompanied antioxidants are insufficient to prevent tissue damage. Therefore, it is vital to try to counteract this with natural supplementary food sources before damage occurs [16].

The World Health Organization (WHO) defines vinegar as a “liquid for human consumption produced by a two-stage fermentation process, first alcohol and then acid fermentation, from suitable raw materials containing starch or sugar” [17]. Vinegar produced from fruits with a high carbohydrate content, such as grapes, apples, mulberries, and hawthorn, undergo fermentation by yeast and bacteria, and the resulting alcohol becomes acetic acid. Since ancient times, vinegar has traditionally been utilized as a flavoring and preservative in various foods and in the treatment of some diseases [18,19]. It is known that vinegar derived from numerous fruits has several compounds such as amino acids, vitamins, minerals, organic acids, and antioxidants. According to these compounds, vinegar has many functional properties, such as antioxidative, antidiabetic, immune-enhancing, and hypocholesterolemic properties [20]. However, the composition of vinegar can be changed based on the production method and raw materials used. Conventional thermal pasteurization and sterilization are the most prevalent methods for inactivating and preserving microorganisms in foods for extended periods. In thermal applications, as the temperature rises, undesirable changes occur, affecting the food’s nutritional value, flavor, and sensory qualities [21,22]. Nevertheless, with the ultrasound technique, among the novel non-thermal technologies increasingly utilized today, the product’s loss of flavor and taste is minimal. In contrast, the product’s nutritional value is enhanced [23,24,25,26,27]. Ultrasound treatment and thermal pasteurization positively affect the antioxidant and phenolic content of fruit, fruit juices, and vinegar [28,29,30,31]. The main components and bioactive compounds of hawthorn vinegar have been revealed in many studies. It has been reported that the total amount of antioxidants in traditional hawthorn vinegar fermentation methods have high phenolic and antioxidant capacities [32]. Additionally, in one study, the fermentation of vinegar increased the number of citric acid and acetic acid compounds [6]. Also, the phenolic components that can be obtained from hawthorn vinegar include gallic acid, caffeic acid, and catechin, all of which can be obtained in large quantities compared to other fruit vinegar. Epicatechin, vanillic acids, ellagic acids, and other hawthorn fruit phenolic components were reported in another research study [33]; however, in this study, these compounds were found to be less prevalent than those discovered by Ozdemir et al. [6]. In the present research study, it was expected that thermal pasteurization and ultrasound treatment would enhance the antioxidant properties of vinegar produced using conventional techniques.

Hawthorn fruit can accelerate metabolism by improving fat burning. It has been reported that hawthorn promotes low-density lipoprotein receptors, thereby decreasing cholesterol [34]. Nevertheless, there have been limited studies on hawthorn vinegar and its effect on health. Researchers have reported that the phenolic compounds of hawthorn vinegar can improve the clinical and biochemical symptoms of atherosclerosis [35]. Hypercholesterolemia and hyperlipidemia are the noteworthy risk factors associated with liver disorders, atherosclerosis, and stroke. The current medications used to treat these complications are incapable of repairing the oxidative stress damage caused by the disease [36]. In this context, there has been a shift towards natural foods, and hawthorn fruit is a superior option; it can also reduce blood cholesterol levels, regulate blood glucose levels, and provide antioxidants, and its efficacy in preventing cardiovascular disease has recently become an essential component in nutrition and nutraceuticals [37]. Our study aimed to examine the efficacy of hawthorn vinegar derived from using thermal pasteurization and ultrasound treatment methods, improve the antioxidant capacity of the fruit, and clarify the effects of the vinegar’s functional properties on the liver and plasma antioxidant parameters, lipid profiles, and immunohistochemical changes in the liver of adult rats.

## 2. Materials and Methods

### 2.1. Animals and Experimental Design

Fifty-six female adult Wistar-Albino rats weighing 230–270 g and aged 6–7 months were housed in Tekirdag Namik Kemal University’s (Turkey) animal housing room at a constant temperature (25 ± 1 °C) for 12 h with fixed darkness and light cycles, and the humidity was maintained 50% to 60%. The animals were fed ad libitum with a standard rodent pellet chow and mains water. All animal experiments were carried out after obtaining local ethical guidelines, and the Ethical Committee approved the experiments conducted at Tekirdag Namik Kemal University. (Approval No. T2021-609). The rats were randomly divided into the 7 groups listed below. Each group hosted 8 rats, and the study involved 1 control group and 6 experimental groups. Hawthorn vinegar (HV) was given to the rats via oral gavage every day at the same hour. The groups were designed as shown below.

Group: Control (no oral HV).Group: HVN0.5 (0.5 mL/kg of untreated hawthorn vinegar consumed orally).Group: HVN1 (1 mL /kg of untreated hawthorn vinegar consumed orally).Group: HVP0.5 (0.5 mL/kg of thermal-pasteurized hawthorn vinegar consumed orally).Group: HVP1 (1 mL/kg of thermal-pasteurized hawthorn vinegar consumed orally).Group: HVU0.5 (0.5 mL/kg of ultrasound-treated hawthorn vinegar consumed orally).Group: HVU1 (1 mL/kg of ultrasound-treated hawthorn vinegar consumed orally).

Before the study, the animals were fed with standard pellets for a 1-week period in order to acclimate them to the environment. Subsequently, the animals were subjected to a care regimen lasting 45 days, during which the experimental groups received hawthorn vinegar via oral gavage as a supplementary component to the pellet, adhering to the predetermined dosage. The animals were sacrificed at the end of the experiment to collect the samples.

### 2.2. Obtaining Vinegar Material

Each trial used 5 kg of hawthorn fruit (*Crataegus tanacetifolia*) as the raw material, which was supplied from Turkey/Bursa. The vinegar was produced using conventional vinegar manufacturing methods. The process involved the maceration of hawthorn fruits in order to extract their essence. The jar contained a 50% mixture of hawthorn and distilled water. Additionally, 5% sugar and 0.5% chickpeas were added to the mixture. The hawthorn mix was placed in sterile jars, and each jar was inoculated with Saccharomyces cerevisiae (3%) to ensure the ethanol fermentation stage. Fermentation lids were closed, and the jars were left at 24–25 °C for 28 days for alcohol fermentation. The fermented product was then transferred into a new sterile jar and inoculated with 10% of sharp vinegar as a source of natural acetic acid culture, subsequently maintained at 28 °C for up to 60 days to obtain a low content (0.5% to 1%) of ethanol. After undergoing filtration, the vinegar was transferred into another container, rendering it prepared for utilization in the treatment phase.

### 2.3. Processing of Vinegar Group Treatments

Vinegar production was carried out by modifying the previous method [38]. The obtained vinegar was processed via 3 methods:Sample without any treatment (groups HVN0.5 and HVN1).Thermal pasteurization of samples in a jar in a water bath at 65 °C for 30 min (groups HVP0.5 and HVP1).The samples were treated with the Response Surface Method and Ultrasound and the application of temperature and time with the best bioactive components as a result of optimization (groups HVU0.5 and HVU1).

### 2.4. Determination of Vinegar’s Bioactive Compounds

The samples’ total phenolic contents (TPCs) were detected using the Folin–Ciocalteu method [39]. All analyses were performed in triplicate. Absorbance was measured using a UV-VIS spectrophotometer (SP-UV/VIS-300SRB, Spectrum Instruments, Victoria, Australia) at a wavelength of 765 nm. The total flavonoid content (TFC) was determined according to the colorimetric technique [40]. The concentrations were calculated colorimetrically using a UV spectrophotometer (Spectrum Instrument, SP-UV/VIS-300SRB, Victoria, Australia) at 510 nm. The ascorbic acid content of the samples was calculated via AOAC 961.27 vitamin preparation and ascorbic acid 2.6 dichlorophenol indophenol-titrimetric method [41]. The antioxidant capacity of the hawthorn vinegar was determined using the CUPRAC method (Cu (II) ion-reducing antioxidant capacity) previously described by the authors of [42]. The DPPH scavenging activity method was used as described by the authors of [43] to determine the antioxidant activity of the samples. 

### 2.5. Blood Samples

The blood samples were collected from puncturing the heart under isoflurane anesthesia at the end of the study and stored in K_3_ EDTA tubes. The blood samples were centrifuged on the same day at 3000 rpm for 10 min to obtain the plasma, and then the plasma was transferred to microtubes. The samples were stored at −80 °C until the day of analysis.

### 2.6. Tissue Sample Preparation

The liver tissue samples intended for ELISA were cut into pieces of appropriate size and weighed before homogenization. The tissues were placed in PBS (pH 7.4) and adjusted to be tissue weight (g): PBS (mL) = 1:9 volume. Then, homogenates were prepared via homogenization using a homogenizer (Interlab, İstanbul, Turkey). The tissue homogenates were centrifuged at 5000× *g* for 5 min, and the supernatants were transferred into microtubes.

Prior to being subjected to immunohistochemistry analysis, the liver tissues were fixed in 10% formalin for 48 h. After fixation, the samples were processed for routine histological protocols and embedded in paraffin. The sections were taken at 4 μm and stained with immunohistochemical staining.

### 2.7. Antioxidant Analyses

The SOD, MDA, and CAT antioxidant parameters of the plasma and hepatic tissues were determined uisng the Enzyme-linked immunosorbent assay (ELISA) method. the samples were measured using a microplate reader (Agilent, Biotek Epoch, Santa Clara, California, USA) Rat SOD Cat No. 201-11-0169, Rat MDA Cat No. 201-11-016, Rat CAT Cat No. 201-11-5106 using commercial ELISA kits (Sun Redbio Hotechology Company, Shanghai, China). A total of 56 animals (8 for each group) were used for those assays. Specimens were placed in microplate wells coated with antioxidant antibodies. The biotinylated antioxidant antibody was added and bound to antioxidants in the sample. It was then bound to the biotinylated antioxidant antibody by adding streptavidin–HRP to the wells. The microplate was incubated for 1 h at 37 °C. After incubation, unbound streptavidin–HRP was removed during washing. Then, substrate solutions were added, and color change was observed in proportion to the amount of antioxidants. The reaction was terminated by adding an acidic stop solution, and absorbance was measured at 450 nm to determine optical density (OD value). Then, the standard curve was constructed by plotting the average OD. The calculated results are shown in units of nmol/mL (MDA) and ng/mL (CAT and SOD).

### 2.8. Biochemical Parameters

Total cholesterol, triglyceride (Ref No: C20T5 and Ref No: TG381, and BEN Biochemical Enterprise, Milan, Italy), HDL and LDL (Ref No: HD320 and LDL348 BEN Biochemical Enterprise, Milan, Italy), and AST and ALT (Ref No: 80025 and 80027 Biolabo, Maizy, France) levels were measured by using spectrophotometric and colorimetric methods. Absorbances were read using a microplate reader (Agilent, Biotek Epoch, Santa Clara, California, USA). The results were computed according to the manufacturer’s instructions. 

### 2.9. Immunohistochemistry Method

The streptavidin–biotin peroxidase complex (strepABC) method was applied to investigate CAT and Mn-SOD immunoreactivity in the liver tissue. The sections were collected on adhesive slides. Following deparaffinization and rehydration, the sections were processed in citrate buffer solution (pH 6.0) for 10 min in a microwave oven at 700 watts and incubated in 3% hydrogen peroxide (H_2_O_2_) for 10 min. In order to inhibit nonspecific bindings, the sections were blocked in an Ultra V Block (Thermo Scientific, Ultravision Large Volume Detection System Anti-Polyvalent, HRP, TA-125-UB, Deutsch, Germany) for 10 min at room temperature. Afterward, the sections were incubated with CAT primary antibody 6 (EPR1928Y, ab76110, Abcam, Cambridge, MA, USA, 1/200 dilution) and Mn-SOD primary antibody (Santa Cruz Biotechnology, Inc., Dallas, Texas USA. B-1: sc-133254, diluted at a rate of 1/100 dilution) in a humid environment at an ambient temperature for 1 h. A Seconder antibody was applied on the sections for 30 min (Thermo Scientific, Ultravision Large Volume Detection System Anti-Polyvalent, HRP, TP-125-BN, Deutsch, Germany), Then, Streptavidin Peroxidase (Thermo Scientific, TS – 125 – HR, Deutsch, Germany) was applied for 30 min. The 3,3′-Diaminobenzidine tetrahydrochloride (DAB) was used as a chromogen for 10 min. Finally, the sections were counterstained with Mayer’s hematoxylin. Negative controls were performed without the primary antibody. The sections were evaluated using a microscope (Olympus BX51, Tokyo, Japan). The immunoreactivity of CAT and MN-SOD were scored. Immunoreactive cells were categorized as having negative, mild, moderate, and intensive immunoreactivity.

### 2.10. Statistical Analyses

Statistical analyses were performed using SPSS (Version 20.0; Chicago, IL, USA). Data were examined for normality distribution and variance homogeneity assumptions (Shapiro-wilk test). If normally distributed, a one-way ANOVA test was applied, and the differences between groups were analyzed using Tukey’s post hoc test. Differences were considered significant at *p* < 0.05, and the means and standard errors were calculated. Nonparametric tests were also used as the data did not provide normal assumptions. Therefore, the differences between the groups were analyzed using the Kruskal–Wallis and Mann–Whitney U tests. Once again, differences were considered significant at *p* < 0.05, and the median values (minimum–maximum) were calculated.

## 3. Results

Hawthorn fruit vinegar (HFV) is a rich source of flavonoid antioxidants. The ultrasound treatment method yielded the highest values with respect to the total amount of phenolic and flavonoid compounds, and DPPH and CUPRAC amounts are given in Table 1. It was observed that the bioactive components of the HVU sample increased compared to the HVN sample. However, it was determined that the bioactive components of the HVN sample decreased during the thermal pasteurization process. As a result, it was determined that ultrasound treatment increased the bioactive components of hawthorn vinegar. This result shows similarities with the literature, as many studies have shown that ultrasound treatments have similar advantages over thermal pasteurization [24,29,31]. However, the antioxidant contexts derived from using the thermal pasteurization processing method (DPHH: 54.86%; CUPRAC: 60.22%; Flavonoid: 13.18 mgCE/100 mL; Total Phenolic: 104.22 mgGAE/100 mL) were lower than those derived from using normal vinegar processing (DPHH: 57.39%; CUPRAC: 63.55%; Flavonoid: 14.22 mgCE/100 mL; Total Phenolic: 110.58 mgGAE/100 mL).

We measured the enzymatic activities of CAT and SOD and the content of MDA in the plasma and liver homogenates of the rats to investigate the antioxidant capacity of Hawthorn vinegar, and the results are presented in Table 2. The results showed that the CAT levels in the plasma (37.51 ± 2.90) and liver tissues (66.10 ± 1.94) of the ultrasound treatment group, 1 mL/kg (HVU1), were the highest compared to the control and other treatment groups, although this result was non-significant. Additionally, the liver SOD level was increased in group HVU0.5 (17.24 ± 0.62) compared to the control group (16.34 ± 0.38) nonsignificantly. Also, the highest liver SOD level value was observed in group HVU1 compared to the control and group HVU0.5 (*p* > 0.05; 18.24 ± 0.35). There were no statistical differences in the MDA levels of the plasma and liver tissues (*p* > 0.05). The liver SOD, CAT, and MDA parameters are also shown in Figure 1, Figure 2 and Figure 3. 

The lipid profiles (total cholesterol, triglyceride, HDL, and LDL) and AST and ALT levels are displayed in Table 3. The biochemical characteristics showed us that the HVP0.5 group had the lowest total cholesterol level among all groups, although this result was statistically not significant (*p* > 0.05; 59.05 ± 3.17). Moreover, it was observed that this metric was lower in the groups treated via ultrasound (HVU0.5: 64.49 ± 4.23 and HVU1: 67.83 ± 3.99) and thermal pasteurization (HVP0.5: 59.05 ± 3.17 and HVP1: 63.36 ± 6.64) compared to the control group (86.34 ± 9.51), although this result was non-significant. Additionally, the highest HDL value was found in the HVU1 cohort but was not statistically significant (*p* > 0.05; 34.00 ± 1.22). On the other hand, LDL levels were significantly lower in groups HVN0.5 (18.20 ± 3.80), HVN1 (20.17 ± 4.83), HVP0.5 (10.91 ± 3.11), HVP1 (10.06 ± 2.98), and HVU1 (13.69 ± 3.10) compared to the control group (48.56 ± 11.75) (*p* < 0.05). Furthermore, the plasma triglyceride levels were not statistically different (*p* > 0.05). Upon examining the hepatic enzymes, there was no difference in ALT values between the groups, and the HVU1 group (84.86 ± 16.95) had the highest AST level compared to the control group (68.79 ± 6.01), although this difference was not statistically significant.

Through our immunohistochemical analysis, it was determined that the CAT and SOD expression intensities of the control, HVN, and HVP groups were less intense than the HVU group in both the 0.5 and 1 mL/kg concentrations of all groups. Mild CAT and SOD expression were observed in the hepatocytes of the control group (Figure 4a and Figure 5a). It was observed that intense CAT and SOD expressions were observed specifically around the blood vessels of all groups (Figure 4 and Figure 5). While mild CAT expression was detected in the hepatocytes of the HVN group (0.5 mL/kg), moderate expression was noted in the HVN group (1 mL/kg), (Figure 4b,e). Moderate CAT intensity was detected in the hepatocytes of the HVP group (Figure 4c,f). Also, intensive expression was detected in the HVU group (0.5 and 1 mL/kg) (Figure 4d,g). Mild SOD expression was observed in the hepatocytes of the HVN group (0.5 mL/kg) (Figure 5b), and intensive expression was observed in the HVN (1 mg/kg) group (Figure 5e). In the HVP groups, moderate expression was observed in the 0.5 mL/kg subgroup, and intensive expression was noted in the 1ml/kg subgroup (Figure 5f). Intensive SOD expression was noted in both HVU groups (0.5 mL/kg and 1 mL/kg) (Figure 5d,g).

## 4. Discussion

Globally, diabetes mellitus, cardiovascular disease, cancer, and Alzheimer’s disease are on the rise, posing a grave risk to public health. Diet quality has been shown to play a crucial role in averting these metabolic disorders. As a consequence of oxidative stress, dietary nutrition leads to potential cellular damage. The presence of complex endogenous defense systems that increase free radical production and decrease antioxidant protection complicates the measurement of oxidative stress status [44]. Consequently, it is feasible to investigate the consumption of antioxidants as a biomarker of oxidative stress by measuring the decrease or increase in antioxidant levels [45]. At this point, consuming foods rich in antioxidants and incorporating more of these foods into our diets is essential. In this regard, hawthorn fruit is an excellent superfood choice due to its components, and it is also one of the earliest used medicinal plants and is extensively distributed throughout China and Europe [46]. In addition, in many cultures, for thousands of years, vinegar from fruit or plants has been used for medicinal treatment as well as nutritional purposes. Researchers have reported that vinegar and its processing methods are important for good nutrition. There have been several experimental and human studies about the effects of vinegar on health; however, more work needs to be carried out to identify the optimal doses of vinegar. The current study provides evidence of the antioxidant efficiency of hawthorn vinegar produced via several methods, namely ultrasound processing and thermal pasteurization. In addition, the findings reported in this study suggest that the highest antioxidant activity was found in the vinegar produced via ultrasound processing. 

*Crataegus* plants have in vitro and in vivo antioxidant properties; they can scavenge superoxide anions, hydroxyl radicals, and hydrogen peroxides and also inhibit lipid peroxidation [47]. Hawthorn fruit is a traditional fruit in Europe, Asia, and China. Over the last few decades, food- and health-based scientific insights have brough increasing attention to the beneficial potential of this food. However, there have been limited studies on hawthorn fruit and hawthorn extracts. Feng et al. [48] found that MDA levels were significantly reduced in rats treated orally with hawthorn fruit acid (HFA). The total antioxidant capacity in the serum and liver of the rats treated with HFA was substantially elevated, as were the activities of the CAT, GSH-px, and SOD enzymes. Similar to this study, in the current study, liver SOD level was found to be higher in group HVU0.5 than in the control group (*p* > 0.05; 16.34 ± 0.38 and 17.24 ± 0.62, respectively). In addition, the liver SOD level in group HVU1 was higher than that in the control and HVU0.5 groups (*p* > 0.05; 16.34 ± 0.38, 17.24 ± 0.62, and 18.24 ± 0.35, respectively). Also, although not significant, there were increases in plasma and liver CAT values in group HVU1 compared to all other groups. However, there were no statistical differences in plasma and liver MDA levels among all groups. When the hepatic tissue and blood plasma samples were compared in terms of their antioxidant parameters, the liver had a higher concentration. Ultrasound treatment has been found to have a beneficial impact on the levels of antioxidants and phenolic compounds in many commodities, including fruit, fruit juices, and vinegar. In the literature, it has been shown that using the ultrasound method in food processing results in the minor degradation of the bioactive components and nutritional characteristics of the goods. Nevertheless, upon reviewing the existing research, it becomes apparent that the ultrasonic investigations available to date have mostly focused on enhancing the quality of vinegar [28]. In this regard, based on the present study, it can be asserted that the ultrasound-assisted processing of hawthorn vinegar improves the product’s antioxidant compounds and thereby improves liver activity and health. 

It has been demonstrated that vinegar, commonly used as a condiment, also has some medical applications. Acetic acid is the primary ingredient in vinegar. Other constituents include anthocyanin, flavonols, vitamins, mineral ions, amino acids, and nonvolatile organic acids [49]. In animal studies, vinegar has exhibited various effects, including enhancing glycogen replenishment, preventing hypertension, and reducing serum total cholesterol and triglyceride levels. It is possible to use vinegar to treat the biochemical risk factors of atherosclerosis, given that vinegar is a safe, widely available, and inexpensive substance. Patients with a high cardiovascular risk respond favorably to the metabolic effects of hawthorn vinegar [50]. In their study, Kadas et al. [35] indicated that hawthorn vinegar consumption decreased body weight, blood pressure, serum glucose, HbA1c, cholesterol, LDL, and triglyceride levels. Similarly, their study revealed a significant rise in HDL and a decline in total cholesterol/HDL. All of these data indicate that vinegar likely possesses a protective effect. 

Hawthorn fruit and hawthorn fruit extracts have demonstrated numerous health benefits, including cardioprotective, hypotensive, and hypocholesterolemic properties. Zhang et al. [51] studied rodents fed a diet containing 30% polyunsaturated canola oil supplemented with 2% hawthorn fruit powder. They reported an increase in serum tocopherol levels by 18–20% compared to the control group animals. As previously mentioned, Kadas et al. [35] proved that hawthorn vinegar supplementation can decrease body weight, serum glucose, cholesterol, LDL, and triglyceride levels. Also, they reported that hyperlipidemia is a prevalent risk factor for the development of atherosclerosis and stroke. These conditions are commonly characterized by raised levels of total cholesterol and/or triglycerides, as well as reduced HDL levels. They implied that a part of the mechanism for the cardioprotective effects of hawthorn fruit may also involve the direct protection of LDL from oxidation or indirect protection by maintaining the concentration of tocopherol in LDL. In addition, a study on rats with HFD-induced hyperlipidemia revealed that the TC and TG levels in their HFA-treated groups decreased progressively over the course of the treatment. These results indicate that the administration of HFA can substantially enhance lipid levels in rats [48]. Regarding our findings, while statistically insignificant, it was observed that the total plasma cholesterol and HDL values were increased in all vinegar groups compared to the control, except group HVU0,5. This may be due to the sample size. On the other hand, the LDL values significantly decreased in all groups compared to the control (*p* < 0.05; 48.56 ± 11.75, 18.20 ± 3.80, 20.17 ± 4.83, 10.91 ± 3.11, 10.06 ± 2.98, and 13.69 ± 3.10 for the control, HVN0.5, HVN1, HVP0.5, HVP1, and HVU1 groups, respectively). However, no significant differences between the groups regarding triglyceride levels and liver enzyme activities were observed. According to our results, vinegar likely has a protective effect on health.

Due to their essential health benefits, consumers frequently select products derived from fruits abundant in phenolic compounds. Hawthorn fruit is rich in organic acids, vitamins, and minerals, as well as phenolic compounds with antioxidant activity and other bioactive compounds [52]. As a result of the bioactive compounds in the hawthorn fruit used in our study, the vinegar made from this fruit had a very high nutritional value. In the analysis of the vinegar employed in our study, the total phenolic and flavonoid compound antioxidant DPPH and CUPRAC ratios were determined to be greatest after using the ultrasonography technique. The authors of a study on tangerine juice stated that the implementation of ultrasound technology is more effective with respect to total antioxidants and food safety than alternative methods [53]. As preservation alternatives for fruit juices and vinegar, ultrasound technology has yielded positive results in maintaining bioactive compound concentration and the physicochemical, microbiological, and sensory qualities of the products [54]. The consumption of fruit vinegar with an extended shelf life and more antioxidant characteristics is preferred by the general public.

The liver, as the organ that is primarily responsible for detoxification and protein synthesis, is involved in a multitude of enzymatic processes. Assessments of liver health can be facilitated by analyzing the concentrations of hepatic enzymes and proteins in the bloodstream. Likewise, heightened concentrations of AST, ALT, ALP, and GGT are indicative of hepatic damage. Specifically, ALT and AST parameters can be measured for cholesterol metabolism. Because these enzymes are located in intracellularly in the liver, if there is liver tissue damage, the levels of these enzymes in the blood will increase. In a study conducted on rats fed a high-fat diet (HFD) and given hawthorn, AST and ALT levels were found to be relatively high in the HFD experimental group; however, there was no change in hepatic enzyme levels in the hawthorn group. Similarly, in our study, there was no statistically significant difference in the AST and ALT levels. It has been suggested that vinegar, even if it is produced via several methods, has no adverse effects on the cells of vital organs and cholesterol metabolism. 

In our investigation, we conducted an immunohistochemistry analysis of the rats’ liver tissues. We observed the presence of CAT intensity in hepatocytes. Furthermore, we observed a significant expression of both CAT and SOD in the HVU groups at doses of 0.5 and 1 mL/kg. It was observed that the groups related to the ultrasound method had more satisfactory antioxidant activity in their liver tissues. In a study similar to our work, it was revealed that hawthorn fruit acid increased the levels of antioxidant enzymes (SOD, CAT) in the liver tissues of hyperlipidemic rats [48].

## 5. Conclusions

The rising global human population is threatening biodiversity, and numerous major illnesses have detrimental effects on human and animal health. Nutrition and stress management are vital for improving health and quality of life. In this context, individuals are being advised to consume natural phytochemicals and antioxidant supplements in order to prevent chronic disorders. Hawthorn fruit, which contains natural phytochemicals, has been shown to possess notable antioxidant capabilities and has exhibited efficacy in reducing body lipids The vinegar derived from this particular fruit, as well as the use of certain techniques to produce the vinegar, has been shown to be capable of enhancing the antioxidative properties of one’s diet. As such, several studies have demonstrated the efficacy of fruit vinegar, such as hawthorn vinegar, in the prevention of free radical generation.

The findings of this study indicated that the application of ultrasound technology resulted in a greater enhancement of the vinegar’s antioxidant and hypolipidemic effects. Also, it was found that the administration of a larger dose (1 mL/kg) of vinegar and the utilization of the ultrasound technique yielded more favorable outcomes compared to the other experimental groups. As a result, we suggest that hawthorn vinegar may have therapeutic effects that could enhance homeostasis and quality of life.

## Figures and Tables

**Figure 1 nutrients-15-03933-f001:**
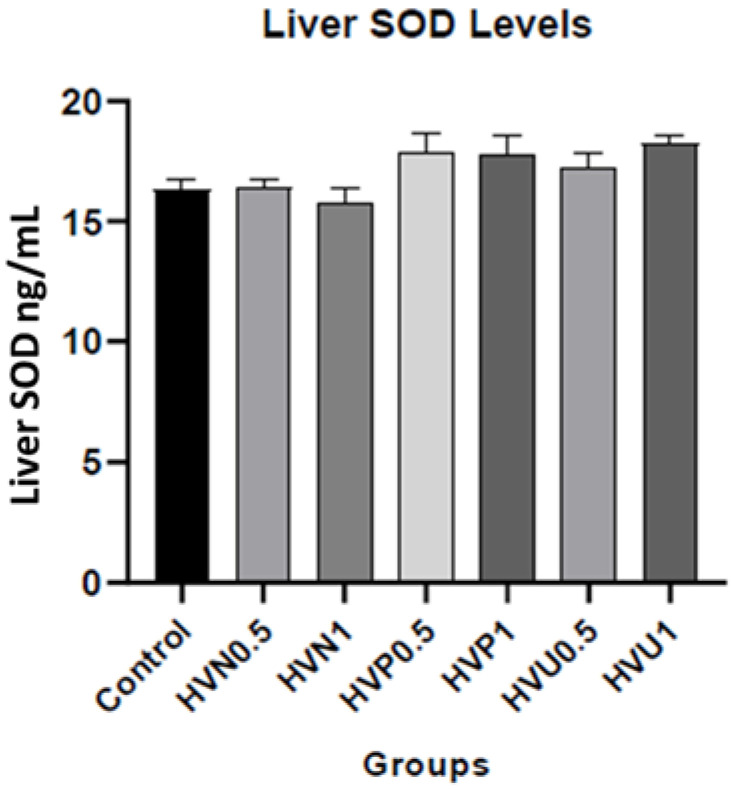
Effects of hawthorn vinegar on liver SOD concentrations in the control and experimental groups. Groups: Control; HVN0.5 (untreated hawthorn vinegar group, 0.5 mL/kg); HVN1 (untreated hawthorn vinegar group, 0.5 mL/kg); HVP0.5 (thermal-pasteurized hawthorn vinegar, 0.5 mL/kg); HVP1 (thermal-pasteurized hawthorn vinegar, 1 mL/kg); HVU0.5 (ultrasound-treated hawthorn vinegar, 0.5 mL/kg); HVU1 (ultrasound-treated hawthorn vinegar, 1 mL/kg). *p* > 0.05; all experimental groups versus control.

**Figure 2 nutrients-15-03933-f002:**
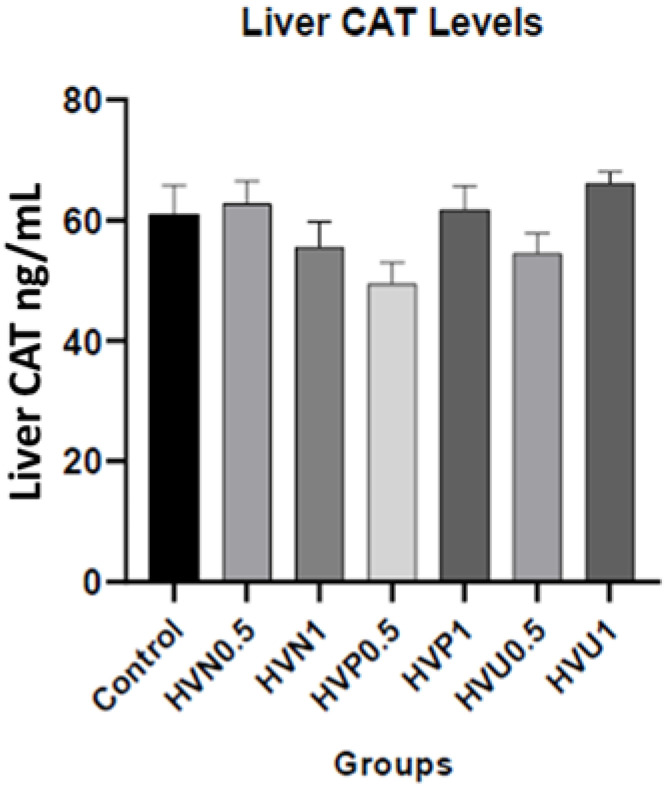
Effects of hawthorn vinegar on liver CAT concentrations in the control and experimental groups. Groups: Control; HVN0.5 (untreated hawthorn vinegar group, 0.5 mL/kg); HVN1 (untreated hawthorn vinegar group, 1 mL/kg); HVP0.5 (thermal-pasteurized hawthorn vinegar group, 0.5 mL/kg); HVP1 (thermal-pasteurized hawthorn vinegar group, 1 mL/kg); HVU0.5 (hawthorn vinegar ultrasound-treated group, 0.5 mL/kg); HVU1 (hawthorn vinegar ultrasound-treated group, 1 mL/kg).

**Figure 3 nutrients-15-03933-f003:**
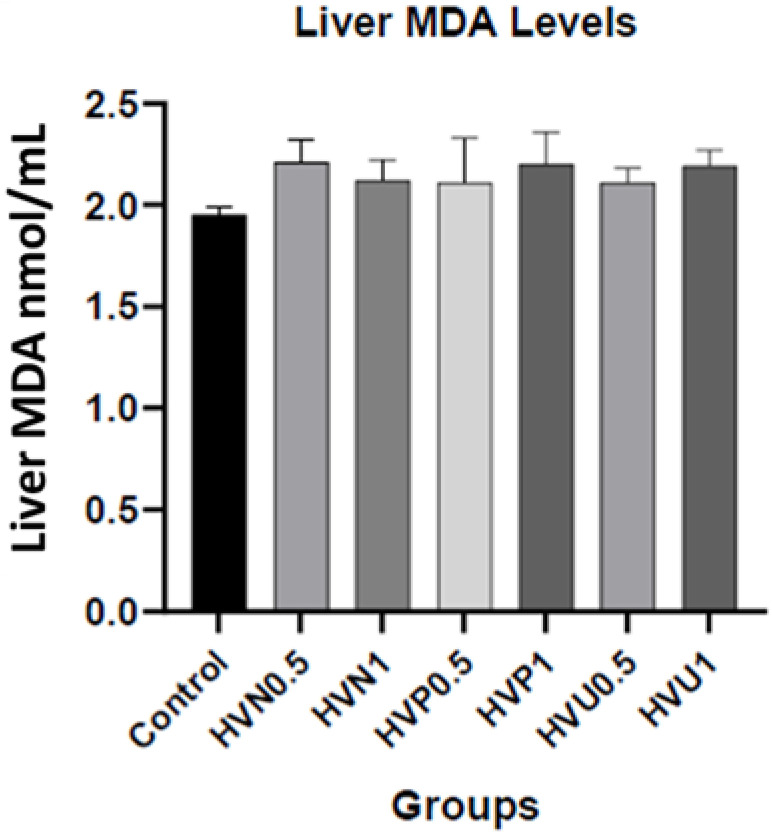
Effects of hawthorn vinegar on liver MDA concentrations in the control and experimental groups. Groups: Control; HVN0.5 (untreated hawthorn vinegar group, 0.5 mL/kg); HVN1 (untreated hawthorn vinegar group, 1 mL/kg); HVP0.5 (thermal-pasteurized hawthorn vinegar group, 0.5 mL/kg); HVP1 (thermal-pasteurized hawthorn vinegar group, 1 mL/kg); HVU0.5 (hawthorn vinegar ultrasound-treated group, 0.5 mL/kg); HVU1 (hawthorn vinegar ultrasound-treated group, 1 mL/kg).

**Figure 4 nutrients-15-03933-f004:**
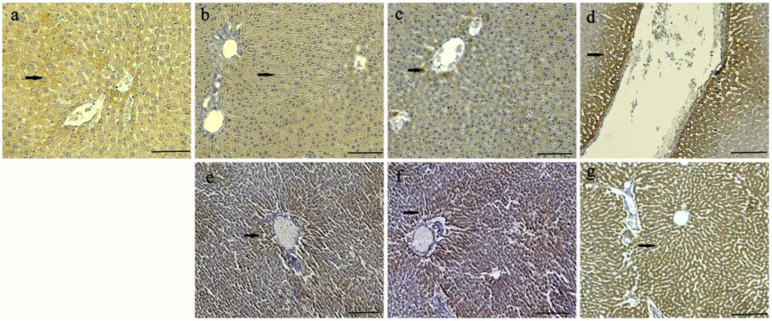
Catalase (CAT) expression in the liver of control and experimental groups. (**a**) Control group; (**b**) HVN0.5 group; (**e**) HVN1 group; (**c**) HVP0.5 group; (**f**) HVP1 group; (**d**) HVU0.5 group; (**g**) HVU1 group. Arrows: hepatocytes. Immunohistochemical staining, bars (**a**–**d**) = 200 μm and bars (**e**–**g**) = 100 μm.

**Figure 5 nutrients-15-03933-f005:**
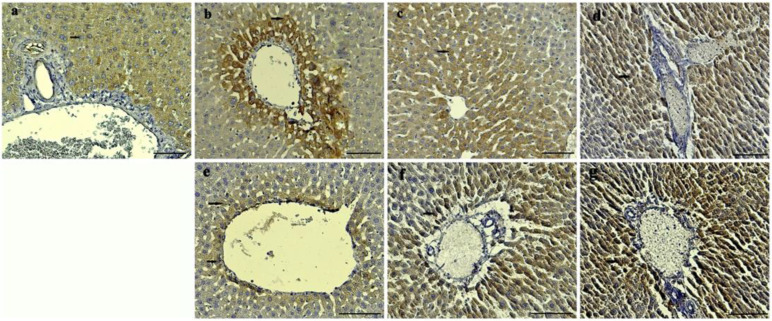
Superoxide dismutase (SOD) expression in the liver of rats. (**a**) Control group; (**b**) HVN0.5 group; (**e**) HVN1 group; (**c**) HVP0.5 group; (**f**) HVP1 group; (**d**) HVU0.5 group; (**g**) HVU1 group. Arrows: hepatocytes. Immunohistochemical staining, bars = 100 μm.

**Table 1 nutrients-15-03933-t001:** Hawthorn vinegar antioxidant content.

Samples	Total Phenolic Compound (TPC) (mg GAE/100 mL)	Total Flavonoid Compound (TFC) (mg CE/100 mL)	Ascorbic Acid (mg/100 mL)	Antioxidant DPPH (% inhibition)	Antioxidant CUPRAC (% inhibition)
Ultrasound-treated (HVU)	116.99	15.89	3.97	62.35	67.39
Normal (HVN)	110.58	14.22	4.22	57.39	63.55
Thermal Pasteurization (HVP)	104.22	13.18	3.45	54.86	60.22

DPPH: 2,2-Diphenyl-1-picrylhydrazyl, CUPRAC: Cupric Reducing Antioxidant Capacity.

**Table 2 nutrients-15-03933-t002:** Effects of hawthorn vinegar administration on antioxidant parameters in rats. All data are presented as the mean ± SE (n:8).

Antioxidant Parameters	Control	Groups					
		HVN0.5	HVN1	HVP0.5	HVP1	HVU0.5	HVU1
Plasma MDA (nmol/mL)	1.41 ± 0.07	1.48 ± 0.08	1.65 ± 0.08	1.60 ± 0.07	1.62 ± 0.02	1.67 ± 0.03	1.57 ± 0.05
Plasma CAT (ng/mL)	39.51 ± 3.18	37.21 ± 1.96	37.89 ± 1.44	42.42 ± 1.66	40.71 ± 0.94	40.94 ± 2.06	37.51 ± 2.90
Plasma SOD (ng/mL)	12.36 ± 0.80	14.62 ± 0.64	11.16 ± 0.23 *	13.64 ± 1.06	14.08 ± 0.40	13.18 ± 0.91	13.30 ± 0.62
Liver MDA (nmol/mL)	1.95 ± 0.04	2.21 ± 0.11	2.12 ± 0.10	2.11 ± 0.22	2.20 ± 0.16	2.11 ± 0.07	2.19 ± 0.08
Liver CAT (ng/mL)	61.01 ± 4.73	62.72 ± 3.78	55.53 ± 4.22	49.42 ± 3.57	61.77 ± 3.91	54.48 ± 3.35	66.10 ± 1.94
Liver SOD (ng/mL)	16.34 ± 0.38	16.38 ± 0.39	15.72 ± 0.69	17.82 ± 0.80	17.80 ± 0.78	17.24 ± 0.62	18.24 ± 0.35

HVN0.5: Untreated Hawthorn Vinegar Group (0.5 mL/kg); HVN1: Untreated Hawthorn Vinegar Group (1 mL/kg); HVP0.5: Thermal-pasteurized Hawthorn Vinegar Group (0.5 mL/kg); HVP1: Thermal-pasteurized Hawthorn Vinegar Group (1 mL/kg); HVU0.5: Ultrasound-treated Hawthorn Vinegar Group (0.5 mL/kg); HVU1: Ultrasound-treated Hawthorn Vinegar Group (1 mL/kg). All data are presented as the mean ± SE. Means within a row with different superscripts (*) differ significantly at *p* < 0.05.

**Table 3 nutrients-15-03933-t003:** The effects of various hawthorn vinegar treatments on the biochemical parameters of the rats. All data are presented as the mean ± SE (n:8).

Biochemical Parameters	Control	Groups					
		HVN0.5	HVN1	HVP0.5	HVP1	HVU0.5	HVU1
Total Cholesterol (mg/dL)	86.34 ± 9.51	72.75 ± 6.69	60.69 ± 6.47	59.05 ± 3.17	63.36 ± 6.64	64.49 ± 4.23	67.83 ± 3.99
HDL (mg/dL)	18.36 ± 3.45	35.24 ± 5.08	28.54 ± 5.58	28.54 ± 4.87	32.26 ± 4.44	15.38 ± 3.46	34.00 ± 1.22
LDL (mg/dL)	48.56 ± 11.75	18.20 ± 3.80 ^a^	20.17 ± 4.83 ^b^	10.91 ± 3.11 ^c^	10.06 ± 2.98 ^d^	28.99 ± 3.98	13.69 ± 3.10 ^e^
Triglyceride (mg/dL)	97.12 ± 3.65	96.56 ± 4.28	89.30 ± 3.06	98.02 ± 5.95	105.17 ± 7.07	100.59 ± 5.86	100.70 ± 6.23
ALT (IU/L)	29.98 ± 2.37	32.65 ± 2.43	32.30 ± 2.45	31.43 ± 5.86	31.43 ± 2.26	32.65 ± 2.35	23.75 ± 3.13
AST (IU/L)	68.79 ± 6.01	59.51 ± 8.89	53.60 ± 8.83	76.65 ± 1.90	54.65 ± 6.46	53.77 ± 4.56	84.86 ± 16.95

HVN0.5: Untreated Hawthorn Vinegar Group (0.5 mL/kg); HVN1: Untreated Hawthorn Vinegar Group (1 mL/kg); HVP0.5: Thermal-pasteurized Hawthorn Vinegar Group (0.5 mL/kg); HVP1: Thermal-pasteurized Hawthorn Vinegar Group (1 mL/kg); HVU0.5: Ultrasound-treated Hawthorn Vinegar Group (0.5 mL/kg); HVU1: Ultrasound-treated Hawthorn Vinegar Group (1 mL/kg). All data are presented as the mean ± SE. Means within a row with different superscripts (^a–e^) differ significantly at *p* < 0.05. ^a^: Control versus HVN0.5; ^b^: Control versus HVN1 group; ^c^: Control versus HVP0.5; ^d^: Control versus HVP1. ^e^: Control versus HVU1 group.

## Data Availability

The data presented in this study are available on request from the corresponding author.

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
