# Peer review of "Ultrasound-Treated and Thermal-Pasteurized Hawthorn Vinegar: Antioxidant and Lipid Profiles in Rats"

_nutrients, 2023, doi:10.3390/nu15183933_

Round 1

Reviewer 1 Report

Dear authors,

the present study is an interesting study not only for the scientific community but also for consumers and a healthy diet support.

Please highlight in abstract and introduction the novelty of the study.

Regarding the section Determination of Vinegar’s Bioactive Compounds and Table 1, please compare with available literature studies and elaborate a bit this section.

Please draw a schematic representation of the study.

Minor spelling

Author Response

Thank you for your enlightening comments. Changes and revisions made are given in the attached file below. 

'' Please see the attachment.

Reviewer 2 Report

The present manuscript reports an in vivo study on the health-related effects of vinegars made from hawthorn fruit. The authors focused mainly on two different methods of vinegar preservation: pasteurization and ultrasound treatment. As a whole, the authors showed that the treatments using ultrasound-treated vinegar led to better liver antioxidant status and blood lipid profiles in rats compared to pasteurized vinegar. Despite the interesting findings, I have several notes for the improvement of the manuscript prior to publication:

-       The title could be arranged in a better manner. The phrase “various hawthorn vinegar methods” is not pertinent. The methods should refer to a certain process, such as production method or preservation method. In addition, since there are only 2 methods investigated in this study, the word “various” should be changed. I would suggest the title to be “the effects of ultrasound treated- and pasteurized-hawthorn vinegar on antioxidant status and lipid profiles in rats”. The last word “rats” should be in plural.

-       Many readers are not familiar with hawthorn fruit, including me. I would suggest the authors to describe hawthorn fruit in a more detailed manner in the introduction session, including its chemical/nutritional content and how to make hawthorn vinegar. The microbiological and chemical processes behind hawthorn vinegar fermentation should be explained thoroughly and clearly in a concise manner.

-       Please explain the logic behind this study. Why did you decide to carry out this study? Is hawthorn vinegar a common food product in Turkey?

-       In the methodology, please describe the hawthorn vinegar fermentation in a more detailed manner. Was there any starter added? What is the purpose of adding chickpeas? Please present the quantity of chickpeas in gram instead of its number.

-       Line 99-104: Group, not grup. I would prefer changing the phrase “hawthorn vinegar normal” into “untreated hawthorn vinegar”.

-       There are serious problems with all the pictures since no statistical analysis was shown In the graphs. Please provide Anova and post-hoc analysis for each graph using the pertinent method to show any significant difference among samples. In addition, please add some information to the figure legends so that each figure can be observed and understood independently on the the manuscript.

-       The conclusion does not reflect the study. Please refer to your objectives and reformulate the conclusion.

Author Response

Thank you for your enlightening comments. Changes and revisions made are given in the attachment file below.

'Please see the attachment.'

Round 2

Reviewer 2 Report

The revised manuscript has included all the solicited revision.